# Bilateral Multiple Tarsal Coalitions (Talonavicular and Talocalcaneal Coalitions) with Recurrent Ankle Sprain in an Adolescent

**DOI:** 10.3390/children9010100

**Published:** 2022-01-12

**Authors:** Chaemoon Lim, Yong-Yeon Chu

**Affiliations:** Department of Orthopedic Surgery, Jeju National University Hospital, Jeju 63241, Korea; yy-chu@hanmail.net

**Keywords:** tarsal coalition, ankle instability, adolescent

## Abstract

Tarsal coalition is defined as an abnormal bony, cartilaginous, or fibrous union of two or more tarsal bones. The incidence of tarsal coalition is approximately 2% in the general population. Talocalcaneal and calcaneonavicular coalitions are the most common. The talonavicular coalition is a rare entity with an incidence of approximately 1.3% among patients with tarsal coalitions. We present a case of a 12-year-old girl who had talonavicular and talocalcaneal coalitions associated with a recurrent ankle sprain. The talonavicular coalition was asymptomatic, and the talocalcaneal coalition was the cause of ankle pain and recurrent sprain. Surgical resection of the talocalcaneal coalition led to successful clinical and functional outcomes. In conclusion, the possibility of multiple tarsal coalitions should be considered in tarsal coalition patients, and the talocalcaneal coalition should be considered as a differential diagnosis in an adolescent patient with a recurrent ankle sprain.

## 1. Introduction

A tarsal coalition is defined as an abnormal bony, cartilaginous, or fibrous union of two or more tarsal bones [1]. The incidence of tarsal coalition is approximately 2% in the general population [2]. Talocalcaneal and calcaneonavicular coalitions are the most common types, with incidences of 48.1% and 43.6% among patients with a tarsal coalition, respectively [2]. The talonavicular coalition is a rare entity with an incidence of approximately 1.3% among patients with tarsal coalitions [3]. The incidence of multiple tarsal coalitions is unknown. Although most of the tarsal coalitions are usually asymptomatic, some patients may have a painful bony prominence or ankle sprain during sports or walking [4,5]. We present a case of a 12-year-old girl who had talonavicular and talocalcaneal coalitions associated with a recurrent ankle sprain. This is the first reported case of a rare combination of tarsal coalitions successfully treated with surgical correction.

## 2. Case Report

A 12-year-old girl was admitted to the outpatient clinic with chief complaints of bilateral ankle pain with an associated recurrent ankle sprain. The patient experienced pain on both the medial and lateral sides of the right ankle for more than three years. She complained of difficulty walking on a bumpy road. Physical examination revealed tenderness on the medial and lateral sides of the ankle joint. The anterior drawer test and inversion and eversion stress test were limited by pain. Plain radiograph revealed bilateral multiple tarsal coalitions including talonavicular and talocalcaneal coalitions in both feet. The American Orthopaedic Foot and Ankle Society (AOFAS) score was 61 and the Oxford Ankle Foot Questionnaires for Child (OxAFQ-C) score was 26 at the initial visit. The talonavicular coalition was synostosis in the right foot and a synchondrosis or syndesmosis in the left foot. C-sign, indicating talocalcaneal coalition, was evident on plain radiographs of both the feet (Figure 1). However, the patient did not complain of any pain in the talonavicular joint regions, and there was no tenderness at the talonavicular joints of both feet. Magnetic resonance imaging (MRI) was performed with the suspicion of ligament injuries as the cause of ankle instability. MRI indicated intact anterior talofibular and calcaneofibular ligaments and fibrocartilaginous talocalcaneal coalitions on both feet (Figure 2). With the provisional diagnosis of a talocalcaneal coalition, conservative treatment (activity modification, anti-inflammatory medication, and physical therapy) was performed for three months. However, the conservative treatment failed, and a surgical correction was performed on the right ankle. A medial incision from the posterior apex of the medial malleolus to the posterior border of the navicular bone was made. The flexor tendon retinaculum and tendon sheet were incised longitudinally. The tibialis posterior tendon was retracted dorsally, and the flexor hallucis longus tendon was retracted plantarly. The bone bridge of the talocalcaneal coalition was identified and excised with an osteotome and burr. After confirming the separation and achieving complete motion of the talocalcaneal joint, auto-graft fat was inserted (Figure 3). Following a subsequent uneventful recovery, the patient remained asymptomatic postoperatively for two years. AOFAS score and OxAFQ-C improved from 61 to 97 and 26 to 3, respectively. The separation of the talocalcaneal coalition was maintained on plain radiographs at two years of follow-up (Figure 4). A less symptomatic talocalcaneal coalition on the left foot is being managed with watchful outpatient follow-up.

## 3. Discussion

The incidence of multiple tarsal coalitions is unknown, and the talonavicular coalition represents 1% of all tarsal coalitions [4]. To our knowledge, this is the first case report of bilateral multiple tarsal coalitions of the talonavicular and talocalcaneal joints with associated lateral ankle instability in an adolescent patient.

The known etiology of congenital tarsal coalitions is a failure of mesenchymal separation between two or more tarsal bones [6,7]. Tarsal coalition is classified on the basis of the morphology of the bridging tissue as a bony (synostosis), chondral (synchondrosis), or fibrous (syndesmosis) coalition [3]. The ossification of talonavicular coalition begins between the ages of 3 and 5 years; that of calcaneonavicular coalition, between the ages of 8 and 12 years; and that of talocalcaneal coalition, between the ages of 12 and 16 years [8]. Most symptomatic tarsal coalitions present as the cartilaginous coalition ossifies. The calcaneonavicular and talocalcaneal coalitions show painful symptoms in the second decade of life as the ossification progresses [9]. However, most of the talonavicular coalitions are asymptomatic or may present with mild midfoot pain [9]. In this case, we speculated that the talonavicular coalition was asymptomatic because of the early ossification, whereas the talocalcaneal coalition caused pain with recurrent ankle sprain as the cartilaginous coalition ossified.

While most of the talocalcaneal coalitions are asymptomatic, the typical clinical presentation of a talocalcaneal coalition in adolescent patients is the restriction of the subtalar joint [7]. The subtalar joint plays an important role in gliding motion during the stance and swing phases in a normal gait [10]. During the stance phase, the subtalar joint accommodates for tibia external rotation. However, when movement at the subtalar joint is restricted due to a talocalcaneal coalition, the tibial external rotation is compensated by the calcaneocuboid or talonavicular joint [3]. Such compensation by these two joints leads to forefoot abduction and adaptive peroneal shortening. This peroneal spasm can cause a peroneal spastic flatfoot in the talocalcaneal coalition [8]. Compensatory actions tend to result in recurrent ankle sprain between the age of 8 and 16 years [7], when subtalar joint motion is restricted. In the current case, we found that the recurrent ankle sprain was caused by the talocalcaneal coalition, rather than ligament injuries. Therefore, in an adolescent with a recurrent ankle sprain, tarsal coalition should be considered in addition to ligament injury.

Talonavicular coalition is an infrequent hindfoot syndrome and represents 1% of all tarsal coalitions [4]. Anderson et al. first reported talonavicular coalitions in 1879, and O’Donoghue described a case of bilateral talonavicular synostosis using radiographs [11]. Genetic mutation and orthopedic anomalies associated with talonavicular coalitions are clinodactyly, symphalangism, the great toe being shorter than the second, clubfoot, a ball-and-socket joint, and metatarsus primus elevation [2,5,12,13]. Most of the talonavicular coalitions are asymptomatic [14]. A medial prominence rather than pain has been reported as the most common clinical finding associated with talonavicular coalitions [4]. Talocalcaneal coalition with minimal symptoms can be managed with conservative treatment, such as soft shoe lifts or a short leg walking cast [11]. Brennan et al. reported a case of talocalcaneal and talonavicular coalitions with associated ankle instability in an adult patient who had successful outcomes following subtalar joint arthrodesis and lateral ligament complex reconstruction only of the talocalcaneal joints, without meddling with the asymptomatic talonavicular joints [9]. 

## 4. Conclusions

To our knowledge, this is the first case report of talonavicular and talocalcaneal coalitions associated with a recurrent ankle sprain in an adolescent. In this rare case of multiple tarsal coalition, surgical resection of the talocalcaneal coalition led to successful clinical and functional outcomes. Moreover, the possibility of multiple tarsal coalitions should be considered in tarsal coalition patients, and the talocalcaneal coalition should be considered as a differential diagnosis in an adolescent patient with a recurrent ankle sprain.

## Figures and Tables

**Figure 1 children-09-00100-f001:**
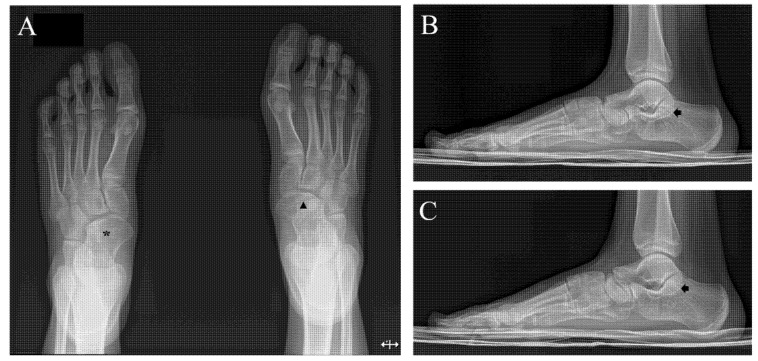
Anteroposterior and lateral radiographs of both feet. (**A**) Anteroposterior radiographs reveal talonavicular coalitions in both feet. The talonavicular coalition was a synostosis (asterisk) in the right foot and a synchondrosis or syndesmosis (arrow head) in the left foot. (**B**) Lateral radiograph shows C-shaped configuration (arrow) of the talocalcaneal coalition at the posterior subtalar joint of the right foot. (**C**) Lateral radiograph shows C-shaped configuration (arrow) of the talocalcaneal coalition at the posterior subtalar joint of the left foot.

**Figure 2 children-09-00100-f002:**
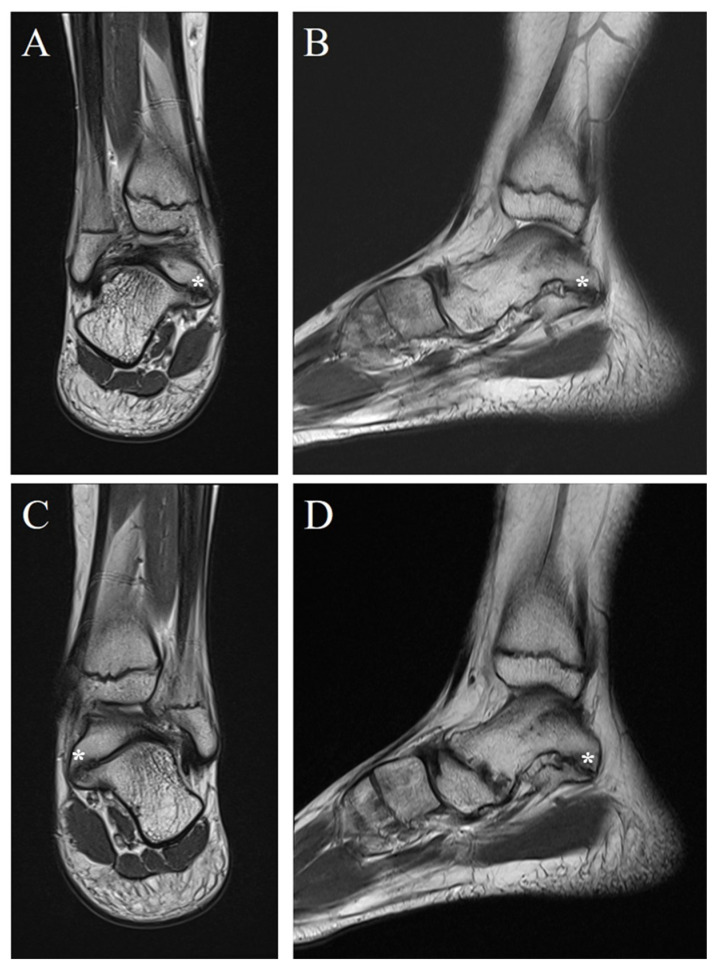
Coronal and sagittal images of magnetic resonance imaging (MRI) of both feet. (**A**,**B**) Coronal and sagittal MRI images of the right foot demonstrate fibrocartilaginous talocalcaneal coalitions at the posterior subtalar joint. (**C**,**D**) Coronal and sagittal MRI images of the left foot demonstrate fibrocartilaginous talocalcaneal coalitions (asterisk) at the posterior subtalar joint.

**Figure 3 children-09-00100-f003:**
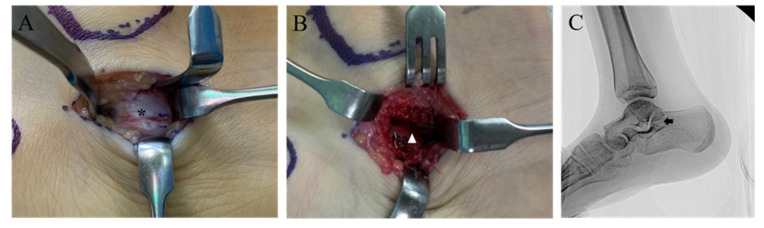
Resection of the talocalcaneal coalition in the right foot. (**A**) The talocalcaneal coalition (asterisk) is seen after retracting the posterior tibial tendon dorsally and flexor hallucis longus tendon plantarly. (**B**) The fibrocartilaginous coalition is excised (arrow head) using an osteotome and burr until the subtalar joint appears. (**C**) After resection of the talocalcaneal coalition, the C-shaped configuration is no longer seen (arrow) on the lateral radiograph.

**Figure 4 children-09-00100-f004:**
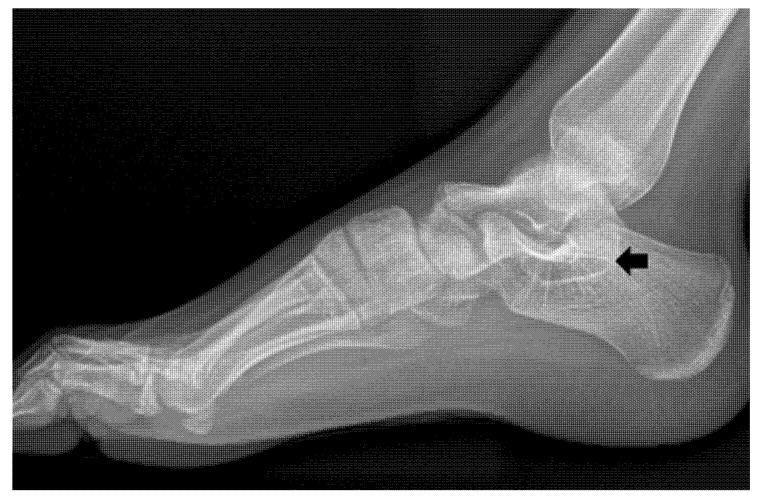
The separation of the talocalcaneal coalition (arrow) is maintained on plain radiographs at two years follow up.

## Data Availability

The data presented in the case report are available on request from the corresponding author.

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
