# Peer review of "Bilateral Multiple Tarsal Coalitions (Talonavicular and Talocalcaneal Coalitions) with Recurrent Ankle Sprain in an Adolescent"

_children, 2022, doi:10.3390/children9010100_

Round 1

Reviewer 1 Report

Page 2 Line 49

Is there any criteria or guideline for the decision of surgical correction of talocalcaneal coalition? Did subject receive any conservative intervention, and what’s the outcome of it?

Page 1 Line 33

What’s the height, weight or BMI of the subject? Is there any other factor which could contribute to the symptoms?

Page 1 Line 39

What’s the score for the sub-categories of AOFAS? Please provide more information regarding the subject’s AOFAS result. Also how did AOFAS change before and after operation?

Page 4 Line 83

  1. What evidence was based to suggest this subject had lateral ankle instability?
  2. Authors claimed it’s the first case report of bilateral multiple tarsal coalitions of the talonavicular and talocalcaneal joints with associated lateral ankle instability in an adolescent patient but it’s not clear what’s the special about this case. The intervention consideration would be different between adolescent and adults? Or is it rare for bilateral multiple tarsal coalitions of the talonavicular and talocalcaneal joints with associated lateral ankle instability to be seen in adolescent? Please clarify the importance of this case.

Page 4 Line 109

Again, what evidence or information was used to conclude the lateral instability? The eversion and inversion stress test were both limited by pain according to the content, and is there any other extra test was performed?

Author Response

Thank you for your review and comment.

We responded with your comment, sincerity.

Response to reviewer 1

Page 2 Line 49

Is there any criteria or guideline for the decision of surgical correction of talocalcaneal coalition? Did subject receive any conservative intervention, and what’s the outcome of it?

-> Most of the talocalcaneal coalitions are asymptomatic. If there are symptom such as, pain or recurrent ankle sprain, conservative treatment begins with activity modification, anti-inflammatory medication, shoe insert or physical therapy. If the conservative treatment for three to six months fails, surgical intervention may be considered. In this case, the patient received conservative treatment for three months, but there was no effect. We added this in manuscript.

Page 1 Line 33

What’s the height, weight or BMI of the subject? Is there any other factor which could contribute to the symptoms?

-> The patient’s height, weight and BMI was 151cm, 45kg and 20kg/m2, respectively. We think that here was not any other factor contribute to the symptoms.

Page 1 Line 39

What’s the score for the sub-categories of AOFAS? Please provide more information regarding the subject’s AOFAS result. Also how did AOFAS change before and after operation?

-> American Orthopaedic Foot and Ankle Society (AOFAS) score is one of the mostly used assessment tool in foot surgery. The questionnaire includes nine items that can be divided into three subscales (pain, function and alignment). Each of the nine items is scored, accumulating to a total score ranging from 0 points (indicating severe pain and impairment) to 100 points (no symptoms or impairment). The patient’s AOFAS score improved from 61 to 97. We attached the preoperative and postoperative AOFAS score in manuscript.

Page 4 Line 83

1. What evidence was based to suggest this subject had lateral ankle instability?

-> We appreciated with your comment. We think a ‘recurrent ankle sprain’ is more suitable than ‘lateral ankle instability’. When the patient first visited out-patient clinic for recurrent ankle sprain, we suspected lateral ankle instability. However, x-ray and MRI confirmed bilateral multiple tarsal coalition (talonavicular and talocalcaneal coalition) and no evidence of ankle ligament injury. Therefore, we considered talocalcaneal coalition to be the cause of the recurrent ankle sprain and treated it. We modified the ‘lateral ankle instability’ to ‘recurrent ankle sprain’ in manuscript.

2. Authors claimed it’s the first case report of bilateral multiple tarsal coalitions of the talonavicular and talocalcaneal joints with associated lateral ankle instability in an adolescent patient but it’s not clear what’s the special about this case. The intervention consideration would be different between adolescent and adults? Or is it rare for bilateral multiple tarsal coalitions of the talonavicular and talocalcaneal joints with associated lateral ankle instability to be seen in adolescent? Please clarify the importance of this case.

-> We appreciated with your comment. The special issue of this case is the rarity of bilateral multiple tarsal coalition (talonavicular and talocalcaneal coalition) with recurrent ankle sprain in adolescent. Talonavicular coalition is rare and multiple tarsal coalition is very rare. Moreover, because most of the tarsal coalition is asymptomatic, it may be difficult to consider tarsal coalition as the cause of recurrent ankle sprain in adolescent. Therefore, tarsal coalition should be considered as a cause of recurrent ankle instability, and the possibility of multiple tarsal coalition should also be considered.

Symptomatic talocalcaneal coalition with recurrent ankle sprain is not rare in adolescent. However, bilateral multiple tarsal coalitions of the talonavicular and talocalcaneal joints with associated recurrent ankle sprain is rare in adolescent.

Like adolescent coalitions, if nonoperative treatment fails, then surgical intervention is considered in adult coalitions. As well as the location of the coalition, the existing advanced arthrosis, and any existing deformity should be considered in adult. Similar to the adolescent, resection can be attempted for talocalcaneal coalitions that do not present with advanced arthrosis or significant hindfoot malalignment. For those patients with advanced arthrosis, more than 50% involvement of the joint hindfoot malalignment, subtalar or triple arthrodesis is recommended. 

Page 4 Line 109

Again, what evidence or information was used to conclude the lateral instability? The eversion and inversion stress test were both limited by pain according to the content, and is there any other extra test was performed?

-> We appreciated with your comment. We appreciated with your comment. We think a ‘recurrent ankle sprain’ is more suitable than ‘lateral ankle instability’. We modified the ‘lateral ankle instability’ to ‘recurrent ankle sprain’ in manuscript.

Reviewer 2 Report

Dear author, I am glad to read cases in the scientific literature, it seems important to me to maintain communication with this format

Author Response

Thank you for your review and comment.